# Real-Time Remote Tele-Mentored Echocardiography: A Systematic Review

**DOI:** 10.3390/medicina56120668

**Published:** 2020-12-02

**Authors:** Alexis Salerno, Diane Kuhn, Rayan El Sibai, Andrea R. Levine, Michael T. McCurdy

**Affiliations:** 1Department of Emergency Medicine, University of Maryland School of Medicine, Baltimore, MD 21201, USA; alexis.salerno@som.umaryland.edu (A.S.); dmkuhn08@gmail.com (D.K.); rayan.sibai@som.umaryland.edu (R.E.S.); 2Division of Pulmonary and Critical Care Medicine, University of Maryland School of Medicine, Baltimore, MD 21201, USA; andrea.levine@som.umaryland.edu

**Keywords:** teleultrasound, clinical ultrasound, echocardiography, critical care, SARS-CoV-2

## Abstract

*Background and Objectives*: Real-time remote tele-mentored echocardiography (RTMUS echo) involves the transmission of clinical ultrasound (CU) cardiac images with direct feedback from a CU expert at a different location. In this review, we summarize the current uses of RTMUS to diagnose and manage cardiovascular dysfunction and discuss expanded and future uses. *Materials and Methods*: We performed a literature search (PubMed and EMBase) to access articles related to RTMUS echo. We reviewed articles for selection using Covidence, a web-based tool for managing systematic reviews and data were extracted using a separate standardized collection form. *Results*: Our search yielded 15 articles. Twelve of these articles demonstrated the feasibility of having a novice sonographer mentored by a tele-expert in obtaining a variety of cardiac ultrasound views. The articles discussed different technological specifications for the RTMUS system, but all showed that adequate images were able to be obtained. Overall, RTMUS echo was found to be a positive intervention that contributed to patient care. *Conclusion*: RTMUS echo allows for rapid access to diagnostic imaging in various clinical settings. RTMUS echo can help in assessing patients that may require a higher level of isolation precautions or in other resource-constrained environments. In the future, identifying the least expensive way to utilize RTMUS echo will be important.

## 1. Introduction

A significant portion of hospitalized patients present with hemodynamic instability [1]. Assessing cardiovascular function is essential to properly manage these patients, and the information garnered from clinical ultrasound (CU) echocardiography routinely alters clinical decisions [2,3]. Unfortunately, in many resource-constrained environments, individuals trained in CU echocardiography may not be immediately available or personnel may be restricted due to a high level of isolation precautions, such as for patients infected with SARS-CoV-2 [4,5].

The use of real-time, remote tele-mentored echocardiography, also termed remote tele-mentored ultrasound echocardiography (RTMUS echo), can provide a useful tool in environments that are resource-constrained or subject to strict isolation precautions by utilizing off-site experts to enhance the clinical information obtained at bedside. RTMUS echo can be used to evaluate left and right ventricular function, valvular function, and to aid in the diagnosis of pathologies such as tamponade. One common way to implement RTMUS echo is through the use of an ultrasound expert at a remote location who provides real-time guidance to novice ultrasound personnel in order to obtain cardiac ultrasound images that can be interpreted to direct further clinical care [6]. 

Although research on the use of RTMUS echo has been increasing in recent years, there is limited literature discussing how and by whom RTMUS echo is being used. We conducted a systematic review with the objective of summarizing the current uses of RTMUS echo to diagnose and manage cardiovascular dysfunction, and based on our findings, to discuss futures uses. 

## 2. Materials and Methods

For this review, we searched the PubMed and EMbase databases for all relevant studies published before 22 July 2020. Please see Appendix A for specific search terms.

Studies were deemed relevant to our systematic review if they met the following criteria: (1) describe RTMUS echo in patients 18 years and older, (2) do not involve the use of augmented reality or robotics, and (3) are written in English. 

The characteristics and results of the selected studies were reviewed, and data were extracted using Covidence, a web-based tool for managing systematic reviews. The data were then recorded in a separate standardized data collection form. The citations were reviewed for relevancy by two members of the study team; if they disagreed, they discussed the articles until reaching a consensus. From the studies that met our selection criteria, we extracted the following data: year published, journal title, authors, article title, study design, objective of study, study setting, number of learners, level of training of the learners, cardiac views performed, technology used, and study results.

## 3. Results

### 3.1. Studies Selected

We identified a total of 753 articles, posters, abstracts, and reviews satisfying the search terms. We excluded 106 duplicate results and 475 irrelevant articles. On closer examination, we excluded 157 studies which did not meet the study criteria. This left 15 studies for inclusion in our review. The flow diagram illustrates the selection process for our review (Figure 1).

### 3.2. Characteristics of Studies

The characteristics of the included studies are summarized in Table 1. The studies ranged widely in terms of questions posed and the specific use of RTMUS echo. The tele-mentors were intensivists, anesthesiologists, emergency physicians, or cardiologists. Twelve of the 15 studies included tele-mentoring of an inexperienced sonographer [8,9,10,11,12,13,14,15,16,17,18,19]. These sonographers included physicians, paramedics, nurses, and non-medically trained individuals who were able to obtain a variety of cardiac ultrasound views.

RTMUS echo was used to evaluate cardiac function and pathology, such as cardiac tamponade or RV strain [12,15]. Four of the articles discussed the use of advanced cardiac evaluations using RTMUS echo, such as E-point septal separation (EPSS) to approximate ejection fraction (EF) [9,13,19], M-mode and Doppler flow measurements to evaluate valvular pathology [13], and inferior vena cava (IVC) respiratory variation to assess signs of fluid responsiveness [14].

Substantial heterogeneity across studies existed among both the types of experts and learners and the outcomes of interest. The varying outcomes included time to image acquisition [11,20], ability to make clinical decisions based on obtained images [10], and degree of correlation of RTMUS echo images findings to other imaging studies or in-person expert sonographer assessments [12,13]. 

The role of technology in RTMUS echo emerged as a recurrent theme highlighted in the reviewed articles. Several different ultrasound machines and communication methods were used among the studies. Please see Table 2 for further details.

## 4. Discussion

### 4.1. An RTMUS Echo Foundation

RTMUS can assist with diagnosing and treating patients in resource-constrained environments [4,6,10,23]. In this review, we specifically looked at the role of RTMUS echo, which can be a powerful tool to identify cardiovascular dysfunction. By using RTMUS echo, providers can evaluate left ventricular function, right ventricular function, and valvular pathologies in critically ill patients. RTMUS can also help to diagnosis emergent conditions such as tamponade, pulmonary embolism, and left ventricular failure. 

As with any technological advancement, RTMUS echo requires a large amount of planning prior to implementation. At present, four integral parts exist in an RTMUS echo program [6]:An ultrasound operator at bedside who can perform images;An ultrasound;A technological platform that can provide active communication between the sonographer and ultrasound expert with simultaneous transmission of ultrasound images and probe location on the patient;An expert ultrasound consultant for interpretation.

### 4.2. RTMUS Echo Training of Novice Users

With little prior education, individuals with different levels of training can adequately obtain ultrasound images [9,10,17]. A large portion of the studies discussed a didactic session prior to initiation of RTMUS. The didactic sessions varied amongst the studies and ranged between 20 and 60 min [8,10,11,12,17,18]. Other studies discussed ultrasound education through RTMUS [9].

### 4.3. RTMUS Technology

Communication must exist between the sonographer and/or ultrasound machine to a remote ultrasound expert. A wide variety of ultrasound machines were used for the studies presented in this analysis, including SonoSite (Fujifilm), LOGIQ S8 (GE Healthcare), and VScan (GE Healthcare) [11,13,16,19,20]. With the increasing prevalence and decreasing cost of small handheld devices such as the Butterfly iQ+, the availability of this technology in remote areas or in low- and middle-income countries is likely to become commonplace. Additionally, small handheld devices may be more feasible in high isolation patients, as they are easier to clean and decrease the risk of device contamination.

### 4.4. RTMUS Interpretation of Images

Effective transmission of high-quality images, which is essential for a functioning RTMUS echo program, was accomplished either through low-cost RTMUS transmission systems [24] or with images/video from publicly available apps such as FaceTime from a 3G or higher cell phone [18]. In general, when ultrasound videos were compressed for transmission, authors indicated that a camera capture rate of at least 30 frames per second was preferable in order to ensure adequate resolution. The authors suggested a resolution of at least 640 × 480 pixels [20,21].

Prior research indicates that images sent by an RTMUS echo system are noninferior to live ultrasound machine images when adequate technology is used [9,10,19,20,21]. For example, the interpretation of EF from images transmitted via a social network was noninferior and had excellent correlation (0.94; *p* < 0.001) to the EF calculated by the modified Simpson method [19]. RTMUS echo also improves image quality as compared to unsupervised images. One study indicated that non-expert images moved 9% closer to expert quality views by using RTMUS echo [22].

### 4.5. RTMUS Technological Considerations

Matching the level of technological support (e.g., US machine, image transmission platform) with the clinical environment is important to ensure a cost-effective program. For example, in a pre-hospital or mass casualty setting, in which identification of active bleeding, pneumothorax, or tamponade may be the highest priority, mild distortions due to video capture rate or resolution may be clinically insignificant. Furthermore, in low-resource settings, tradeoffs between quality of image and the availability of ultrasonography may focus on using the least expensive technology able to answer the clinical question at hand.

### 4.6. RTMUS Echo Use in SARS-CoV-2 and Other High Isolation Areas

The SARS-CoV-2 pandemic has strained almost every aspect of society and particularly, health care systems. The scientific community is adapting and learning how to contend with SARS-CoV-2. Though knowledge of the effects of the pandemic on the respiratory and cardiovascular systems is still in its infancy, research and progress are exponentially increasing. Several imaging techniques are available to diagnose, understand, and clinically manage the effects of SARS-CoV-2. These techniques include chest X-ray, computed tomography (CT) of the chest, and CU. Chest radiographic findings of pulmonary disease are usually absent early in the disease and sensitivity may be only 69% [25,26]. CT scans offer the advantage of providing significant information such as estimation of disease severity and evaluation for pulmonary embolism [26]. The American College of Radiology advocates CT scanner decontamination after imaging any patient with suspected SARS-CoV-2, a practice that limits CT scanner availability [27]. Additionally, CT scans require patient transport throughout the hospital, increasing the possibility for infection control failures. Given the volume of SARS-CoV-2 patients, many of whom are critically ill, using CT may be unsafe and impractical. On the contrary, CU can be performed at bedside, with portable devices being left in a patient’s room or within a biocontainment unit, decreasing contamination risk.

CU can be used for cardiac and pulmonary evaluation, severity stratification, and monitoring of patients with suspected SARS-CoV-2 infection. It can be used to identify left ventricular diastolic dysfunction as well as right ventricular dilatation and impairment, findings which have been described in patients with severe SARS-CoV-2 infection [28].

CU is an integral part of the evaluation of a critically ill patient. Most of these patients will have serial CU examinations to look for changes in their EF and to look for signs of fluid tolerance or intolerance. CU operators are in close contact with patients and therefore, at a high risk of being infected. Expert consensus supports judiciously selected imaging techniques to minimize the risk of infectious exposure of health care personnel or other patients [29]. As such, the most experienced operator may not be the ideal operator in every clinical situation. Thus, the role of RTMUS may increase among SARS-CoV-2 patients to retain expert evaluation while minimizing physical interaction with patients.

### 4.7. Future Direction and Research of RTMUS Echo

RTMUS echo is a newer technology with great potential to provide high-level expertise to low-resource settings. In the future, we will need to advance the use of RTMUS echo to become standardized in these settings. 

Most of the articles we reviewed are small studies or case reports. In the future, large scale studies will help to establish the use of RTMUS echo in different clinical settings. Future studies on RTMUS echo must also revisit the tradeoffs between time and cost, on the one hand, and superior data transfer and image resolution, on the other. One step toward addressing these tradeoffs will be to determine a minimally useful image quality and the technology required to achieve this quality. Finally, although cost of technology was briefly addressed in some studies, identifying the least expensive and easiest ways to implement tele-ultrasound, such as with a handheld device or with asynchronous transmission, will be important.

## 5. Conclusions

In our review of the literature, 15 articles addressed the use of RTMUS echo to evaluate cardiac dysfunction. These studies highlighted the feasibility of tele-mentoring novice sonographers to obtain clinically useful images, as well as tele-mentoring for the sake of education and improving comfort with the use of ultrasonography. To decrease provider exposure to high risk areas and conserve limited personal protective equipment, RTMUS echo may have an increasing role in use among patients in high isolation clinical settings such as those needed to manage patients with COVID-19. Staying abreast of the fastest and most cost-effective ways to implement RTMUS echo to produce and transmit quality images for interpretation will help to ensure that ongoing changes in technology are appropriately being leveraged to deliver optimal patient care. 

## Figures and Tables

**Figure 1 medicina-56-00668-f001:**
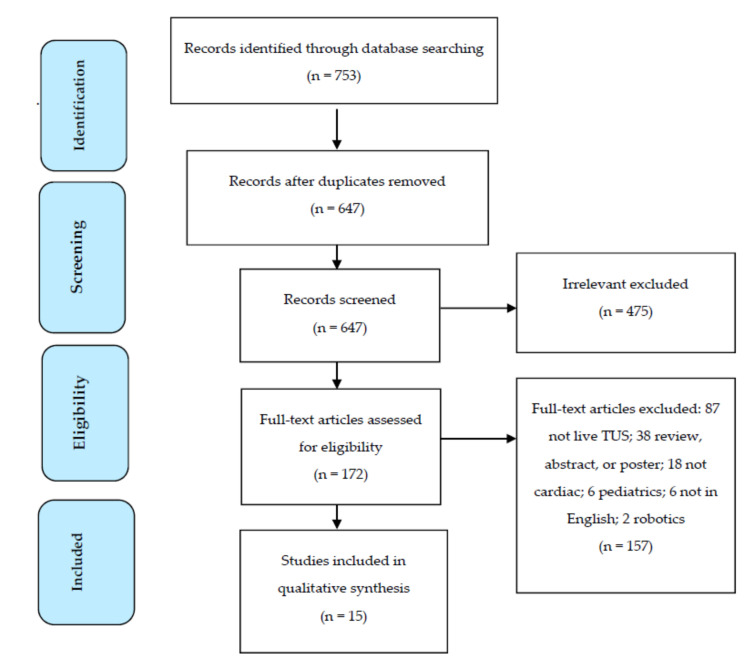
Prisma Flowsheet for Study Selection [7].

**Table 1 medicina-56-00668-t001:** RTMUS Echo Study Features.

Article	Study Design and Setting	Objective	Number and Experience Level of Learners/Sonographers	Number of Patients and Patient Population	Cardiac US Views and Measurements Obtained	Outcome Measures	Results
Afset et al. (1996) [13]	Feasibility study, community hospital, Norway	Evaluate reproducibility and accuracy of RTMUS echo measurements	One novice physician	38 patients with known or suspected heart disease	PSL, A4C; EPSS, Doppler to estimate MR, AR, TR/pulmonic flow	Learner’s measurements compared to expert US examination	No difference between expert and RTMUS assessments of mean M-mode or Doppler variables
Miyashita et al. (2003) [20]	Feasibility study, mobile health vehicle, Japan	Evaluate ability to effectively transmit US images via satellite	Echocardiography specialist	57 patients	Not specified	Image quality and acquisition time compared to US machine images	Average exam time 8.4 min (range, 6.1–10.1). Quality nearly identical to original
Huffer et al. (2004) [21]	Feasibility study, simulated mass casualty, USA	Determine feasibility and diagnostic accuracy of RTMUS during mass casualty	Trained sonographers	10 individuals with known structural cardiac disease and 2 healthy controls	LVEF, RV Strain, RWM, LV size, AV/MV pathology	Technical quality and diagnostic accuracy	Overall average of 95% concordance between two sets of images
Boniface et al. (2011) [8]	Feasibility study, simulated pre-hospital setting, USA	Assess ability of paramedics to obtain adequate views using RTMUS	51 novice paramedics	Healthy volunteer	Subx; PSL if inadequate subx	Adequacy of views (yes/no)	Success rate of 100% to obtain “adequate” views
Otto et al. (2012) [15]	Case report, community hospital, Antarctica	Demonstrate ability of RTMUS to serve as important diagnostic tool in remote environments	One physician with “basic” CU skills	One patient	Subx, PSL, PSS, A4C	No formal analysis	Diagnosis of pericarditis; RTMUS prevented unnecessary transcontinental medical evacuation
Russell et al. (2014) [9]	Prospective, randomized, single-blinded, academic setting, USA	Compare ability to obtain PSLA US views for (1) no, (2) remote, and (3) in-person mentoring	18 novice medical students	75 kg live model	PSL; EPSS	Adequacy and quality of EPSS	No significant difference
Levine et al. (2016) [18]	Feasibility study, academic setting, USA	Determine ability of telemedicine ICU physicians to mentor remote sonographers to obtain US images	11 novice non-physician health care providers	One healthy volunteer	Subx	Compare image quality and ability to make clinical decisions from US machine or RTMUS	Of RTMUS images, 69/77 (90%) were high quality and 74/77 (96%) permitted clinical decision making
Becker et al. (2017) [14]	Case report, tertiary hospital ICU, USA	Evaluate fluid responsiveness	One provider with CU training	One patient	PSL, A4C, IVC	RTMUS exam showed signs of distributive and hypovolemic shock	Patient given fluid with increase in MAP and vasopressors stopped
Kim et al. (2017) [19]	Feasibility study, tertiary hospital ICU, Korea	Determine ability of remote expert to evaluate EF with RTMUS using social network video call	60 novice sonographers	60 patients	PSL, PSS, A4C; EPSS, EF	Compare cardiologist-performed Simpson’s method vs. RTMUS EF and EPSS evaluation	Statistically excellent agreement between two measurements of EF
Robertson et al. (2017) [10]	Feasibility study, community hospital, Haiti	Determine ability of remote tele-intensivist to mentor providers to obtain US images	9 novice non-physician healthcare workers	One healthy volunteer	Subx	Comfort of making clinical decisions based on images, image quality	Tele-intensivist could make clinical decisions with 56/63 (89%) images, of which 57/63 (90%) were high quality
Epstein et al. (2018) [16]	Rural hospital, Uganda	Evaluate ability of physician to detect major US findings after basic training	One physician underwent 5-day training	7 echo studies, not specified # of real-time	Not specified	Assess image utility to make clinical decisions	RTMUS via smartphone for echo image feasible, reducing need for complete echo studies
Douglas et al. (2019) [17]	Feasibility study, pilot cohort and clinical cohort, community ICU, USA	Assess US training (1) effect on non-physician comfort performing TUS and (2) feasibility to improve participant comfort	Pilot cohort: 11 non-physician providersClinical cohort: 5 ICU nurses	Pilot cohort: 1 healthy volunteerClinical cohort: ICU inpatients over 6 weeks	Pilot cohort: SubxClinical cohort: PSL, PSS, Subx	Participant survey of experience and comfort of performing RTMUS	After training, all participants had positive experience and comfortable using RTMUS
Jensen et al. (2019) [22]	Single-blinded cluster randomized control trial, regional ED, Denmark	Investigate image quality of cine-loop recordings of RTMUS vs. non-supervised physician’s vs. experts	10 physicians with prior CU training	44 patients	Subx, PSL, PSS, A4C	Two blinded observers graded cine-loops recorded from all scans	RTMUS images had higher image quality than those by unsupervised physicians
Ramsingh et al. (2019) [11]	Feasibility study, academic setting, USA	Assess anesthesiologist ability to guide remote nonmedical learners to obtain US images	21 novice non-medically trained students	One healthy volunteer	PSL, PSS	Image acquisition time, Quality of Image	Average exam time 8.5 min, 90% cardiac images had ≥3 out of 4 quality rating
Olivieri et al. (2020) [12]	Feasibility study, community ICU, USA	Evaluate ability of RTMUS to approximate CU exam performed by provider	5 novice ICU nurses	20 patients	PSL, PSS, Subxiphoid	Concordance between RTMUS and CU and clinical test	High specificity for all abnormalities

Abbreviations: RTMUS, real-time remote-mentored teleultrasound; PSL, parasternal long axis view; A4C, apical four-chamber; EPSS, end point septal separation; MR, mitral regurgitation; AR, aortic regurgitation; TR, tricuspid regurgitation; US, ultrasound; LV, left ventricle; PE, pulmonary embolism; RWM, regional wall motion abnormality; AV, aortic valve; MV, mitral valve; EP, emergency physician; Subx, subxiphoid view; CU, clinical ultrasound; PSS, parasternal short axis view; MAP, mean arterial pressure; EF, ejection fraction; ED, Emergency Department.

**Table 2 medicina-56-00668-t002:** RTMUS Echo Study Features.

Article Title	US Machine	Recording and Transmission Technology	Capture, Resolution, and Data Transfer Rate	Delay, Distortions, and Cost
Afset et al. (1996) [13]	270 SSA (Toshiba)	Videoconference signals digitized by computer before transmission through digital telecommunication channels using a system called MEGA-NET	Video codec (Philips VCD 2M-G) compressed video signals by 97%. MEGA-NET capacity of 2 Mbps.	Cost of equipment ~USD 34,500 per site. Cost of video conference ~(USD 28/h)
Miyashita et al. (2003) [20]	Dyna View-II SSD 1700 (Aloka)	Remote-controlled camera at exam site, images transmitted using satellite links (JCSAT-1B) as videoconferencing and DICOM images	Meeting system could transmit images of 640 × 480 pixels at an upload rate of 30 fps at best.	RTMUS system cost USD 30,000 and the communication cost~ USD 4 per min.
Huffer et al. (2004) [21]	VISICU, Inc.	MPEG-2 compression technology	Capture rate of 32 fps, needed higher gain than usual.	
Boniface et al. (2011) [8]	Sonosite Micromaxx, M-Turbo (Fujifilm)	Physician communicating with paramedic via two-way radio		
Otto et al. (2012) [15]	Acuson US and TeleRad workstation	Transmitted via McMurdo Station’s T-1 satellite communications link	Data link of 384 Kbps to McMurdo Station’s LAN	
Russell et al. (2014) [9]	Vscan(GE Healthcare)	Mentoring via Google Glass and Google Hangouts		Google Glass estimated at USD 1500
Levine et al. (2016) [18]	SonoSite S-ICU (Fujifilm)	Tele-ICU camera: images captured using Sony camera	Camera had 340° pan, 120° tilt, 18× optical, 12× digital, and 380k pixels	
Becker et al. (2017) [14]	Not provided	No information provided	No information provided	No information provided
Kim et al. (2017) [19]	Logiq S8 (GE Healthcare)	Video call (Kakao face talk) with 4G network using a Galaxy S7 (Samsung)	US machine with 1920 × 1080-pixel LED	
Robertson et al. (2017) [10]	SonoSite M-Turbo (FujiFilm)	Apple MacBook laptop, connected to sonographer in Haiti via an Apple iPhone 5S, both operating Apple’s FaceTime app	Apple iPhone running FaceTime using 4G cellular data network	
Epstein et al. (2018) [16]	Vscan (GE Healthcare)	Cellular phones, commercially available video-chat software, and 3G cellular data network		
Douglas et al. (2019) [17]	SonoSite X-Porte (Fujifilm)	Two-way camera to visualize both US machine and sonographer; remote tele-intensivist used Philips monitoring software		
Jensen et al. (2019) [22]	Vivid S6 (GE Healthcare)	Video grabber (DVI2USB 3.0; Epiphan Video), two web cameras, headset, two laptop computers (on-site and remote)		
Ramsingh et al. (2019) [11]	SonoSite Edge (Fujifilm)	Apple FaceTime and Google Glass, with one-way visual communication and two-way audio communication		
Olivieri et al. (2020) [12]	SonoSite X-Porte (Fujifilm)	Philips audiovisual communication link, Philips monitoring software, and videoconference with camera (Sony EVI-D70)	Camera had 18x lens with horizontal resolution of 470 television lines (TVL)	Facilities and equipment from preexisting tele-ICU

Abbreviations: 3G, third generation; 4G, fourth generation; DICOM, Digital Imaging and Communications in Medicine; EP, emergency physician; fps, frames per second; Kbps, kilobits per second; LAN, local area network; Mbps, megabits per second; TVL, television lines. All monetary values in US dollars.

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
