# Peer review of "Real-Time Remote Tele-Mentored Echocardiography: A Systematic Review"

_medicina, 2020, doi:10.3390/medicina56120668_

Round 1

Reviewer 1 Report

The authors have reviewed 15 relevant published articles about Real-time remote-mentored tele-echocardiography (REMUS echo) which enables one to remotely diagnose cardiovascular dysfunction for patients who are resource-restrained or are restricted due to high level of isolation precautions. 

It is important to have a current knowledge of the developments and implementation of technology in the medicinal field. in this article the authors are gathering the literature of the current and past usages of REMUS echo for diagnostic purposes. It is crucial in pandemics such as COVID-19 where isolation is important, to have high class images of the patient's ultrasound for the expert to diagnose accurately, and at low cost.

On line 13 in Abstract, 'diagnosis' should be 'diagnose'.

Author Response

Thank you for taking the time to review our article. We have changed line 13 to “diagnose.”

Reviewer 2 Report

In this systematic review of real time remote mentored tele-echo articles which found 15 articles , the majority promoting the use of RTMUS echo.

The introduction requires some more information on why it the use of RTMUS echo is vital and strengthen what the gap in the literature is.

I suggest you put the MeSH terms as an appendix and not in the main text.

Why did you choose the inclusion criteria of over 17 years?

What years did you include in your search?

Author Response

Thank you for taking the time to review our article.

  • We have added line 37-38 to show why the use of RTMUS echo is vital, and line 42-43 about literary gap.
  • We have removed the MeSH terms from the methods section and placed them in Appendix A.
  • We chose to include patients older than 17 years because as adult providers we have familiarity in the use of RTMUS in adult populations and are not experience in the use of RTMUS in pediatrics. We have changed this to 18 years and older to clarify.
  • Please see line 44, we searched relevant studies from before July 22,2020.

Reviewer 3 Report

The paper titled “Real-Time Remote-Mentored Tele-echocardiography: A Systematic Review” by Salerno and colleagues is a systematic review of the literature about the current uses of Remotely Tele-Mentored ultrasound (RTMUS) to diagnose and manage cardiovascular dysfunction. Moreover, the authors focused on the use of RTMUS in special clinical settings where patients require a higher level of isolation precautions such as in case of SARS-CoV-2 infection. The article is well-written, quite informative and will be helpful for those who want to learn more about this issue.

However, the authors might consider the following comments:

  • The abbreviation spelled out in the abstract and in the introduction is not appropriate: RTMUS stands for Remotely Tele-Mentored UltraSound.
  • Line 56: “Studies that included patients under age 17…” is a useless sentence due to the fact the authors have already stated that one of the criteria for the systematic review was articles that described RTMUS echo in patients older than 17 years.
  • Although the paper gives the readers the opportunity to select articles that are informative about the issue, the authors should emphasize which are the main echocardiographic parameters that inexperienced sonographers have to evaluate in critical patients.
  • Regarding the paragraph dealing with critical patients with SARS-CoV-2 infection, I really do not know if the evaluation of their lungs should be part of an echocardiography study. Moreover, authors stated that they excluded from the revision studies that imaged structures other than heart or inferior vena cava. Thus we have to consider the paragraph 4.6 something that is not in line with the title of the paper.
  • The paragraph 4.7 about future direction and research of RTMUS Echo should be expanded.

Author Response

Thank you for taking the time to review our article. 

  • Thank you for recognizing this detail. We have changed the title, abstract and introduction to state real time remote tele-mentored echocardiography (RTMUS echo). We have mentioned remote tele-mentored ultrasound echocardiography as an additional recognized term in the introduction but due to the repetitive nature of this term have chosen not to use this full term in the title and abstract.
  • We have deleted the repetitive statement, see line 49.
  • We have added a lines 105-108 to discuss the main echocardiographic parameters used in RTMUS echo.
  • We have removed specific statement about heart and IVC and decided to state echocardiography broadly. Chest xray can be important to look for signs of pulmonary edema due to heart failure, ct is used to look for pulmonary embolism and signs of RV strain. Since these are pathologies involving the cardiovascular system we have decided to keep the first paragraph. We have removed the sentence about ultrasound specific pulmonary findings see line 169.
  • Added lines 184-187.